# Human airway macrophages are metabolically reprogrammed by IFN-γ resulting in glycolysis-dependent functional plasticity

Donal J Cox, Sarah A Connolly, Cilian Ó Maoldomhnaigh, Aenea AI Brugman, Olivia Sandby Thomas, Emily Duffin, Karl M Gogan, Oisin Ó Gallchobhair, Dearbhla M Murphy, Sinead A O'Rourke, Finbarr O'Connell, Parthiban Nadarajan, James J Phelan, Laura E Gleeson, Sharee A Basdeo*[†], Joseph Keane[†]

Trinity Translational Medicine Institute, St James's Hospital, Trinity College Dublin, The University of Dublin, Dublin, Ireland

*For correspondence:
basdeos@tcd.ie

[†]These authors contributed equally to this work

Competing interest: The authors declare that no competing interests exist.

## eLife Assessment

In this **valuable** study, the authors investigate how inflammatory priming and exposure to irradiated *Mycobacterium tuberculosis* or the bacterial endotoxin LPS impact the metabolism of primary human airway macrophages and monocyte-derived macrophages. The work shows that metabolic plasticity is greater in monocyte-derived macrophages than alveolar macrophages, with **solid** experimental methods and overall evidence. The findings are relevant to the field of immunometabolism.

**Abstract** Airway macrophages (AM) are the predominant immune cell in the lung and play a crucial role in preventing infection, making them a target for host directed therapy. Macrophage effector functions are associated with cellular metabolism. A knowledge gap remains in understanding metabolic reprogramming and functional plasticity of distinct human macrophage subpopulations, especially in lung resident AM. We examined tissue-resident AM and monocyte-derived macrophages (MDM; as a model of blood derived macrophages) in their resting state and after priming with IFN-γ or IL-4 to model the Th1/Th2 axis in the lung. Human macrophages, regardless of origin, had a strong induction of glycolysis in response to IFN-γ or upon stimulation. IFN-γ significantly enhanced cellular energetics in both AM and MDM by upregulating both glycolysis and oxidative phosphorylation. Upon stimulation, AM do not decrease oxidative phosphorylation unlike MDM which shift to 'Warburg'-like metabolism. IFN-γ priming promoted cytokine secretion in AM. Blocking glycolysis with 2-deoxyglucose significantly reduced IFN-γ driven cytokine production in AM, indicating that IFN-γ induces functional plasticity in human AM, which is mechanistically mediated by glycolysis. Directly comparing responses between macrophages, AM were more responsive to IFN-γ priming and dependent on glycolysis for cytokine secretion than MDM. Interestingly, TNF production was under the control of glycolysis in AM and not in MDM. MDM exhibited glycolysis-dependent upregulation of HLA-DR and CD40, whereas IFN-γ upregulated HLA-DR and CD40 on AM independently of glycolysis. These data indicate that human AM are functionally plastic and respond to IFN-γ in a manner distinct from MDM. These data provide evidence that human AM are a tractable target for inhalable immunomodulatory therapies for respiratory diseases.

**eLife digest** Inside the human body, immune cells known as macrophages are constantly looking for microbes, cell debris and other potential threats to engulf and digest. If a macrophage detects a microbe, it activates and releases molecules called cytokines, which induce further immune responses that help to eliminate the invader.

The macrophages found in the lungs, known as airway macrophages, defend against pollutants and airborne microbes and are therefore key for maintaining respiratory health. Despite this, previous studies have suggested that airway macrophages are not as good at responding to infections as other types of macrophages.

Certain cytokines can cause macrophages to switch how they generate the chemical energy needed to fuel various processes in the cell. However, it remains unclear if it may be possible to develop therapies that boost airway macrophage activity during infection by modifying how they produce chemical energy.

To investigate, Cox et al. compared how human airway macrophages and macrophages that originate in the blood alter their production of chemical energy in response to cues from the immune system that indicate an infection is present. The experiments showed that exposure to a specific cytokine known as IFN-γ caused both macrophage types to produce more chemical energy using a metabolic process known as glycolysis.

Inhibiting glycolysis induced by IFN-γ had a much bigger effect on the ability of the airway macrophages to produce cytokines than it had on blood macrophages. Furthermore, glycolysis controlled the production of a particular cytokine called TNF in the airway macrophages, but not the blood macrophages.

The findings demonstrate that airway macrophages alter how they produce chemical energy during infections in a different way to blood macrophages. Since TNF is a crucial cytokine for defending against respiratory infections, understanding how it is regulated in the lung could help researchers to develop inhalable therapies to boost its production in patients with respiratory infections that are difficult to treat. The specificity of this approach could ultimately limit side effects compared to therapies that act throughout the body.

## Introduction

Airway macrophages (AM) are the sentinels of the lung and the first responders to respiratory insults such as infection. Despite a large body of evidence indicating that these tissue resident AM have a distinct phenotype and function to peripherally derived macrophages, there remains a significant lack of data regarding human AM function and plasticity in response to infection and their ability to change under the influence of Th1 or Th2 environments. Macrophage function exists on a spectrum of activation states based on tissue residency, ontogeny, cytokine milieu and the plasticity of the macrophage in response to environmental factors (*Huang et al., 2018*; *Ginhoux et al., 2010*). Much of the research has focused on the contribution of metabolic pathways to polarising macrophages into distinct pro-inflammatory or regulatory phenotypes (*Wang et al., 2018*). There is a paucity of data on the role of metabolism in response to Th1 or Th2 microenvironments induced by cytokines such as IFN-γ or IL-4 respectively, in human macrophages, especially in tissue resident macrophages, such as AM. A knowledge gap remains as to whether the tissue resident AM is metabolically and functionally plastic and therefore capable of mounting effective pro-inflammatory responses despite its homeostatic, regulatory tissue resident phenotype.

Plasticity of macrophage function requires metabolic reprogramming (*Vijayan et al., 2019*; *Tannahill et al., 2013*). Since AM play a key role in directing and propagating immune responses and inflammation in the lung, we sought to determine the plasticity of AM and monocyte-derived macrophages (MDM). Using primary human AM and MDM, we modelled Th1 and Th2 microenvironments with the addition of IFN-γ or IL-4, respectively. To further examine the function of IFN-γ or IL-4 primed macrophages, we stimulated cells with the gram-negative bacterial component, lipopolysaccharide (LPS), or whole bacteria, irradiated *Mycobacterium tuberculosis* (Mtb; iH37Rv). Firstly, we assessed the metabolic phenotype of unprimed human AM, or primed with IFN-γ or IL-4. IFN-γ significantly increased the cellular energetics of both human AM and MDM. Furthermore, subsequent stimulation

led to an increase in the extracellular acidification rate (ECAR), a surrogate marker of glycolysis in both macrophages. Therefore, using the glycolytic inhibitor 2-deoxyglucose (2DG) we then examined the mechanistic role of glycolysis in the phenotypic and functional plasticity of both AM and MDM. Herein the functional plasticity is defined as the ability of primed macrophages to differentially alter cytokine production in response to bacterial stimuli whereas the phenotypic plasticity is defined by alterations in surface expression of activation markers.

These data demonstrate that human AM are functionally plastic and respond to IFN-γ or IL-4 differently than MDM. These novel data demonstrate differential metabolic responses within human macrophage subpopulations that are linked with functionality. Furthermore, these data address a knowledge gap in human respiratory innate immunology and provide evidence that the AM is a tractable target to support human respiratory health.

## Results

### IFN-γ induces metabolic reprogramming in both AM and MDM

AM alter their metabolism in response to Mtb (*Gleeson et al., 2016*). Human macrophages also undergo a rapid increase in ECAR early in response to activation (*Ó Maoldomhnaigh et al., 2021a*) and these pathways can be pharmacologically manipulated (*Cox et al., 2020*; *Phelan et al., 2020*). The metabolic and functional plasticity of human AM remains unexplored; however, evidence shows they express both 'M1' and 'M2' markers (*Mitsi et al., 2018*). Murine AM can be reprogrammed through an IFN-γ-dependent mechanism (*Yao et al., 2018*). We therefore sought to examine whether priming human AM with IFN-γ compared with IL-4 or unprimed AM, could influence their metabolic function and response to bacterial stimuli. We stimulated with whole bacteria; Mtb (iH37Rv) or gram-negative cell wall component; LPS. AM were plated in a Seahorse plate and primed with IFN-γ or IL-4 (both 10 ng/ml) for 24 hr or left unprimed. AM ECAR and oxygen consumption rate (OCR) were recorded for 30 min at baseline. AM were then stimulated in the Seahorse XFe24 Analyzer with medium (control), iH37Rv (MOI; 1–10) or LPS (100 ng/ml) and ECAR and OCR were continuously monitored (*Figure 1—figure supplement 1A*&B).

At 150 min post stimulation fold change compared to unprimed unstimulated AM was calculated for ECAR (*Figure 1A*) and OCR (*Figure 1B*). IFN-γ priming significantly increased the ECAR and OCR of unstimulated human AM compared with control or IL-4 primed AM (*Figure 1A and B*). Upon stimulation with iH37Rv or LPS, AM significantly increased ECAR compared to their respective unstimulated controls, regardless of cytokine priming (*Figure 1A*). IFN-γ primed and subsequently stimulated AM exhibited a significantly increased ECAR compared with stimulated control or IL-4 primed AM (*Figure 1A*). IL-4 primed iH37Rv stimulated AM increased ECAR to similar extent as unprimed controls (*Figure 1A*; left). Conversely, IL-4 primed AM stimulated with LPS AM did not increase their ECAR to the same extent as controls (*Figure 1A*; right), suggesting that IL-4 reduces the AM ability to increase ECAR in response to LPS stimulation. IFN-γ significantly increased the OCR of AM in response to stimulation with iH37Rv or LPS, and had enhanced OCR compared with other stimulated controls (*Figure 1B*). These data indicate that priming human AM with IFN-γ increases both glycolytic and oxidative metabolism, which is then further increased upon stimulation.

Since IFN-γ priming increased cellular energetics in the AM at baseline, we calculated percent change in ECAR and OCR from the baseline rate of each group in order to assess if IFN-γ or IL-4 primed AM have altered capacity to change their metabolism in response to stimulation (*Figure 1C and D*). This was carried out to equalise all the primed data sets at baseline before stimulation (*Figure 1—figure supplement 1C and D*). These data indicate that whilst the peak of glycolysis is elevated in IFN-γ primed AM (*Figure 1—figure supplement 1A*), all AM have a similar capacity to increase glycolysis upon stimulation when baseline differences in metabolism were adjusted for the effects of cytokine priming (*Figure 1C*). IFN-γ increased the percent change in OCR of AM in response to both bacterial stimuli compared to the unstimulated IFN-γ primed control (*Figure 1D*). These data indicate that priming AM alters the metabolic baselines of human tissue resident macrophages and not their ability to respond to bacterial stimuli.

In order to compare the metabolic responses of AM with blood-derived macrophages, we next assessed MDM. Human MDM were left unprimed or primed with IFN-γ or IL-4 (both 10 ng/ml). 24 hr after cytokine priming metabolic flux was monitored by recording ECAR and OCR at baseline for

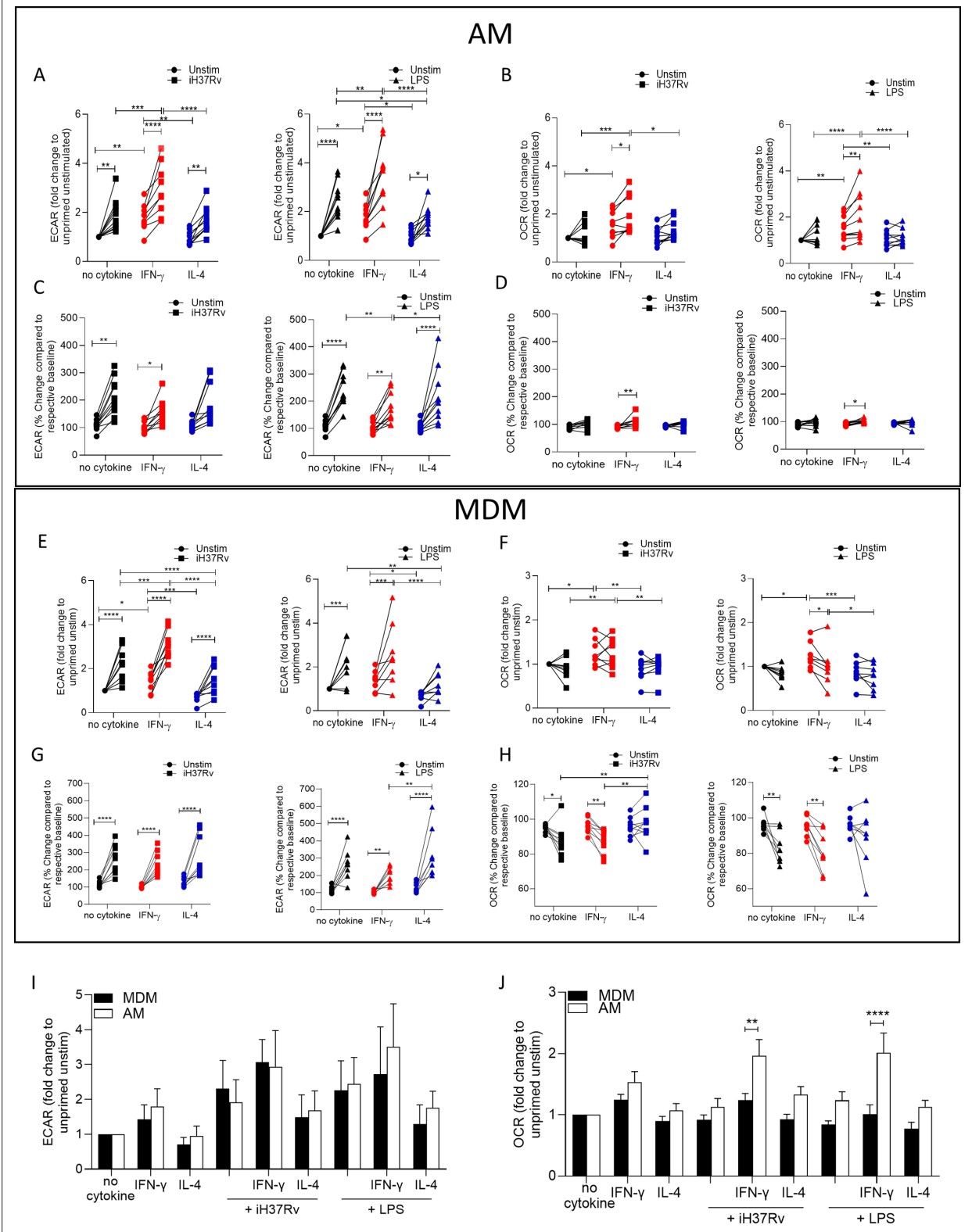

**Figure 1.** IFN-γ increases energetic metabolism in the AM but enhances 'Warburg'-like metabolism in MDM in response to inflammatory stimuli. Human AM (**A–D**) were isolated from bronchoalveolar lavage fluid. PBMC were isolated from buffy coats and MDM (**E–H**) were differentiated and adherence purified for 7 days in 10% human serum. Cells were left unprimed (black) or primed with IFN-γ (red) or IL-4 (blue; both 10 ng/ml) for 24 hr. Baseline measurements of the Extracellular Acidification Rate (ECAR) and the Oxygen Consumption Rate (OCR) were established before AM or MDM were

*Figure 1 continued on next page*

*Figure 1 continued*

stimulated with medium (circle), irradiated Mtb H37Rv (iH37Rv; MOI 1–10; square) or LPS (100 ng/ml; triangle), in the Seahorse XFe24 Analyzer, then monitored at 20 min intervals. At 150 min, post stimulation fold change in ECAR (**A, E, I**) and OCR (**B, F, J**) was analysed, and percentage change (from baseline of the respective treatment group) was also calculated for ECAR (**C, G**) and OCR (**D, H**) at 150 min. A direct comparison of AM (white bar) and MDM (black bar) was also assessed at 150 min (**I, J**). Each linked data point represents the average of technical duplicates for one individual biological donor (MDM; n=8–9, AM; n=9–10). Statistically significant differences were determined using two-way ANOVA with a Tukey (**A–H**) or Bonferroni post-test (**I–J**); *p≤0.05, **p≤0.01, ***p≤0.001, ****p≤0.001.

The online version of this article includes the following figure supplement(s) for figure 1:

**Figure supplement 1.** Human AM and MDM seahorse traces for ECAR and OCR.

30 min. MDM were then stimulated with medium, iH37Rv or LPS and ECAR and OCR was continuously monitored (*Figure 1—figure supplement 1E and F*).

As per previous observations (*Ó Maoldomhnaigh et al., 2021a*; *Ó Maoldomhnaigh et al., 2021b*) a sustained increase in ECAR and a transient decrease in OCR occurred in MDM after stimulation (*Figure 1—figure supplement 1E–H*). At 150 min post stimulation, fold change was calculated compared to unprimed unstimulated MDM (*Figure 1E and F*). IFN-γ priming significantly increased the ECAR and OCR of MDM whereas IL-4 priming significantly reduced the ECAR in the absence of stimulation (*Figure 1E and F*). Stimulation of human MDM with iH37Rv or LPS significantly increased ECAR in all MDM; however, IL-4 primed MDM stimulated with iH37Rv or LPS have significantly reduced ECAR compared with control or IFN-γ primed MDM (*Figure 1E*). IFN-γ primed MDM stimulated with iH37Rv have increased ECAR compared with control MDM (*Figure 1E*).

Similar to AM, IFN-γ primed MDM have increased OCR compared with control or IL-4 primed MDM (*Figure 1F*). In contrast with the AM, stimulation of IFN-γ primed MDM does not further increase OCR however, the elevated OCR in IFN-γ primed MDM remains significantly higher compared to control or IL-4 primed MDM when stimulated with iH37Rv (*Figure 1F*). The percent change in ECAR upon stimulation (from respective baselines) illustrates that all MDM groups significantly increase ECAR from their own baseline in response to stimulation (*Figure 1G*). Interestingly, although IL-4 significantly reduced ECAR in iH37Rv and LPS stimulated MDM compared with unprimed stimulated controls (*Figure 1E*), the IL-4 primed MDM have significantly enhanced capacity to ramp up glycolysis in response to LPS, as evidence by the significantly increased percentage change in LPS stimulated, IL-4 primed MDM compared with IFN-γ primed controls (*Figure 1G*). Control or IFN-γ primed MDM stimulated with either iH37Rv or LPS decreases percentage change in the OCR associated with a stimulation-induced shift to 'Warburg'-like metabolism (*Figure 1H*). This effect is not observed in IL-4 primed MDM, moreover, IL-4 primed MDM stimulated with iH37Rv had significantly elevated percent change in OCR compared with stimulated unprimed or IFN-γ primed MDM (*Figure 1H*). These data indicate that IL-4 priming prevents human MDM utilising 'Warburg'-like metabolism in response to stimulation.

Since AM and MDM had distinct responses to priming and stimulation, we next directly compared the metabolic responses of AM and MDM. AM and MDM had similar levels of ECAR relative to their own unprimed controls, which were both enhanced upon stimulation (*Figure 1I*). The OCR is elevated in the AM compared with the MDM; IFN-γ primed AM exhibit significantly increased OCR compared with MDM in response to stimulation with iH37Rv or LPS (*Figure 1J*).

In summary, human AM upregulate glycolysis early in response to stimulation. IFN-γ significantly promoted cellular energetics (both ECAR and OCR) in unstimulated AM which was further enhanced by stimulation. IFN-γ promotes increased cellular energetics in stimulated human MDM by promoting both glycolysis and oxidative phosphorylation, whilst maintaining the capacity for the cells to shift to 'Warburg'-like metabolism in response to stimulation. IL-4 priming significantly reduced the cellular energetics compared with control or IFN-γ primed MDM. Importantly, IL-4 prevents the drop in OCR occurring in stimulated MDM thereby inhibiting 'Warburg'-like metabolism. IL-4 primed AM had reduced fold change in glycolysis upon stimulation with LPS compared with controls.

## IFN-γ promotes HLA-DR and CD40 markedly more on human MDM than AM whereas IL-4 promoted CD86

Having established that energetic responses are plastic in response to IFN-γ in the AM and that post stimulation energetic responses are different in human macrophage types under Th1 or Th2 priming conditions, we next sought to determine the effect on the plasticity of the macrophage phenotype by

examining expression of activation markers associated with antigen presentation function. Human AM (*Figure 2A, C and E*) and MDM (*Figure 2B, D and F*) were primed with IFN-γ or IL-4 for 24 hr or left unprimed. Macrophages were then stimulated with iH37Rv or LPS. After 24 hr, AM and MDM were analysed by flow cytometry for expression of HLA-DR (*Figure 2A, B and G*), CD40 (*Figure 2C, D and H*), and CD86 (*Figure 2E, F and I*). A sample gating strategy for the analysis is provided (*Figure 2—figure supplement 1*).

IFN-γ significantly increased the expression of HLA-DR compared with control or IL-4 primed unstimulated AM (*Figure 2A*). Stimulation with iH37Rv significantly upregulated HLA-DR, but only in unprimed AM (*Figure 2A*). Similarly, LPS significantly induced HLA-DR in unprimed or IL-4 primed AM but not in IFN-γ primed AM (*Figure 2A*). IFN-γ also significantly increased the expression of HLA-DR compared with control or IL-4 primed MDM (*Figure 2B*). Stimulation of IFN-γ primed MDM with iH37Rv or LPS robustly enhanced the expression of HLA-DR (*Figure 2B*). IFN-γ priming significantly upregulated CD40 expression in unstimulated AM (*Figure 2C*; right). In addition, CD40 was upregulated following iH37Rv or LPS stimulation of AM in all groups assessed with the exception of IFN-γ primed AM stimulated with iH37Rv (*Figure 2C*). IFN-γ increased the expression of the co-stimulatory molecule CD40 in unstimulated MDM (*Figure 2D*). Stimulation of MDM with iH37Rv or LPS significantly increased CD40 expression, with the exception of iH37Rv stimulation in IL-4 primed MDM (*Figure 2D*). Expression of CD86 in response to stimulation with iH37Rv was only upregulated in IL-4 primed AM; however, LPS induced upregulation of CD86 in all AM, with IFN-γ and IL-4 primed AM exhibiting significantly enhanced CD86 expression compared to unprimed control (*Figure 2E*). CD86 expression induced by MDM in response to iH37Rv or LPS was enhanced by priming with either IFN-γ or IL-4, with IL-4 inducing significantly higher expression compared with unprimed or IFN-γ primed MDM (*Figure 2F*).

In order to directly compare human AM and MDM responses to IFN-γ and IL-4, fold change in HLA-DR, CD40 and CD86 was calculated compared to the average of respective unstimulated unprimed controls (*Figure 2G–I*). The human MDM has increased HLA-DR and CD40 expression in response to IFN-γ compared to the human AM (*Figure 2G and H*). This increased expression of HLA-DR and CD40 by MDM, becomes even more profound after stimulation. MDM also have greater expression of CD86 when primed with IL-4 compared to AM, which was again enhanced by stimulation (*Figure 2I*). IFN-γ primed MDM stimulated with iH37Rv also increased expression of CD86 compared to AM (*Figure 2I*).

## MDM are more dependent on glycolysis for the IFN-γ-driven upregulation of HLA-DR and CD40 while AM are not

Since IFN-γ drove glycolysis and the expression of the macrophage activation markers CD40 and HLA-DR in both AM and MDM we wanted to examine if the increased glycolysis was associated with enhanced expression of activation markers expression. Human AM (*Figure 3A, C and E*) and MDM (*Figure 3B, D and F*) were primed with IFN-γ or IL-4 for 24 hr or left unprimed. Macrophages were then treated with the glycolytic inhibitor, 2DG for 1 hr prior to stimulation with iH37Rv or LPS. After 24 hr, AM and MDM were analysed by flow cytometry for expression of HLA-DR (*Figure 3A and B*), CD40 (*Figure 3C and D*), and CD86 (*Figure 3E and F*). 2DG-mediated inhibition of glycolysis following stimulation was confirmed (*Figure 3—figure supplement 1A*).

Inhibiting glycolysis with 2DG did not alter expression of HLA-DR on AM (*Figure 3A*). Interestingly, the increased expression of HLA-DR in IFN-γ primed MDM was dependent on glycolysis in unstimulated and iH37Rv stimulated MDM, however, increased expression of HLA-DR by LPS stimulated IFN-γ primed MDM remained elevated in the presence of 2DG (*Figure 3B*). Expression of CD40 was not affected by 2DG in unstimulated or iH37Rv stimulated AM (*Figure 3C*). Conversely, LPS induced expression of CD40 was significantly inhibited by 2DG in unprimed and IFN-γ primed AM but not in IL-4 primed AM (*Figure 3C*). In contrast, enhanced expression of CD40 in IFN-γ primed MDM in unstimulated or iH37Rv stimulated MDM was significantly reduced with the addition of 2DG, with no effect on the expression of CD40 in LPS stimulated human MDM regardless of cytokine priming (*Figure 3D*). 2DG enhanced expression of CD86 in unstimulated IFN-γ or IL-4 primed AM but did not affect expression in any stimulated AM (*Figure 3E*). Interestingly, when AM were examined in the absence of stimulation, IFN-γ priming significantly increased CD86 (*Figure 3E*). 2DG inhibited the increased expression of CD86 in response to iH37Rv stimulation in IFN-γ or IL-4 primed MDM, but no difference was observed in unstimulated or LPS stimulated MDM (*Figure 3F*).

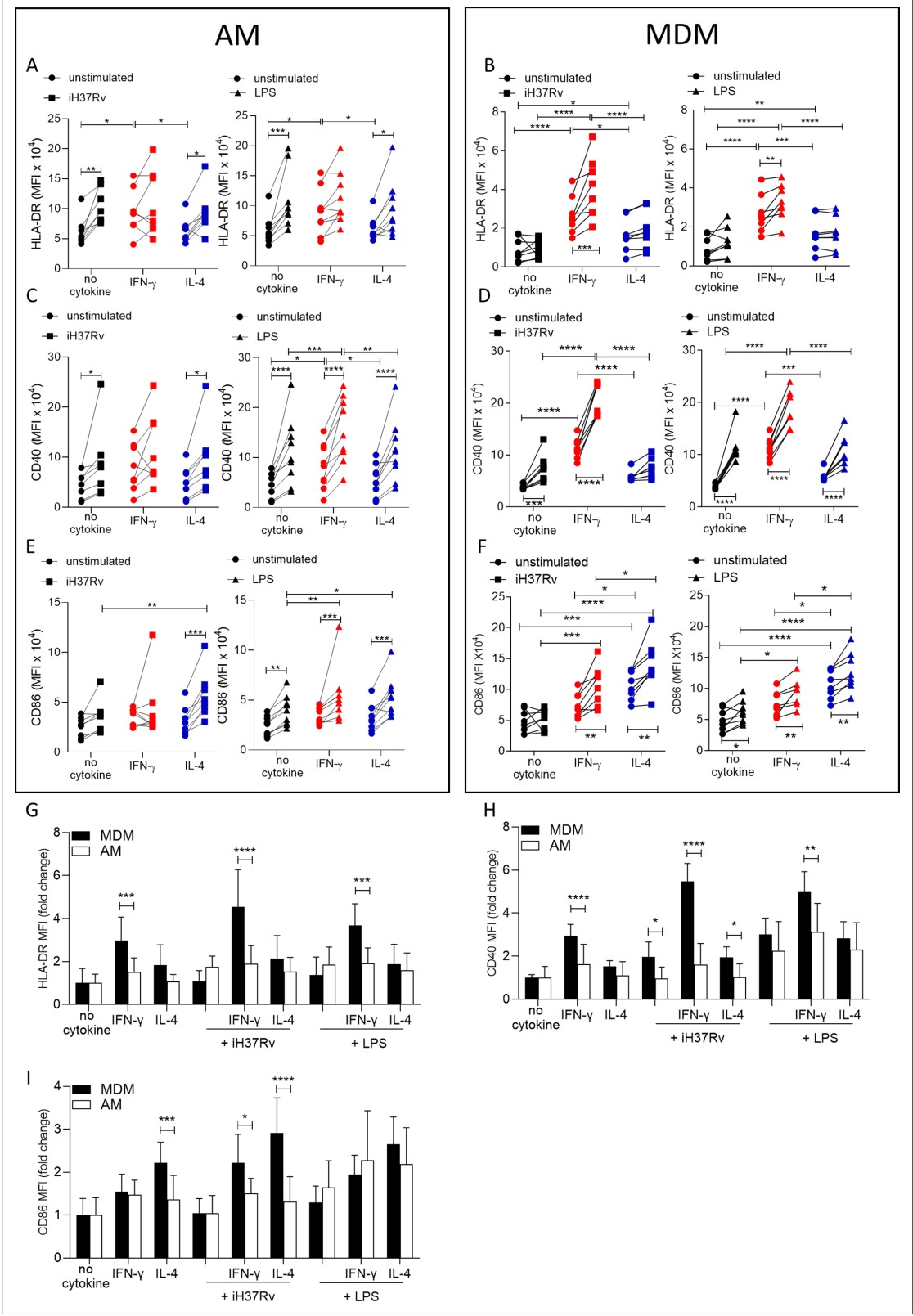

**Figure 2.** IFN-γ boosts activation marker expression on MDM to a greater extent than AM. Human AM (**A, C, E**) isolated from bronchoalveolar lavage fluid. PBMC were isolated from buffy coats and MDM (**B, D, F**) were differentiated and adherence purified for 7 days in 10% human serum. Cells were left unprimed (black) or primed with IFN-γ (red) or IL-4 (blue) (both 10 ng/ml) for 24 hr. AM or MDM were left unstimulated (circle) or stimulated with iH37Rv (MOI 1–10; square) or LPS (100 ng/ml; triangle). After 24 hr, cells were detached from the plates by cooling and gentle scraping and stained

*Figure 2 continued on next page*

*Figure 2 continued*

for HLAR-DR (**A, B**), CD40 (**C, D**), CD86 (**E, F**) and analysed by flow cytometry. Fold change of HLA-DR (**G**), CD40 (**H**) and CD86 (**I**) was calculated for AM (white bar) and MDM (black bar) based on the average of their respective no cytokine controls. Each linked data point represents the average of technical duplicates for one individual biological donor (n=8–9). Statistically significant differences were determined using two-way ANOVA with a Tukey (**A–F**) or Bonferroni post-test (**G–I**); *p≤0.05, **p≤0.01, p***≤0.001, ****p≤0.001.

The online version of this article includes the following figure supplement(s) for figure 2:

**Figure supplement 1.** Supplementary flow cytometry gating strategy.

Cumulatively, these data indicate that IFN-γ upregulates the expression of activation markers more effectively in human MDM than AM. Since these markers are associated with activating T cells during presenting antigen, the ability of IFN-γ or IL-4 primed MDM to process antigen was next assessed, along with the dependency on glycolysis. MDM were primed with IFN-γ or IL-4 for 24 hr or left unprimed. MDM were then treated with 2DG for 1 hr prior to stimulation with DQ-Ovalbumin (500 ng/ml) for 30 min. IL-4 primed MDM had significantly reduced ability to process DQ-Ovalbumin compared with control or IFN-γ primed MDM (*Figure 3—figure supplement 1B*). Treatment of MDM

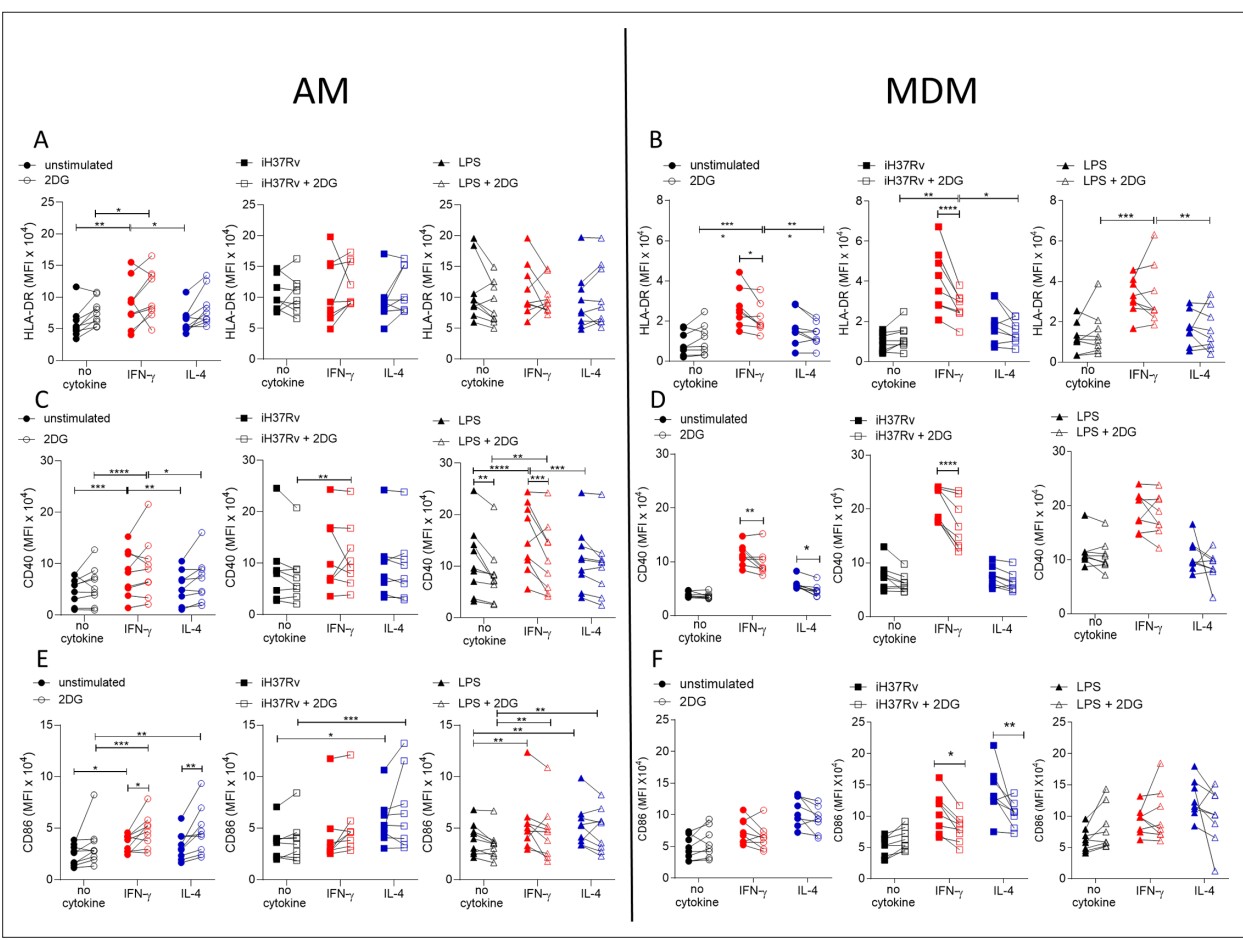

**Figure 3.** Glycolysis is required for IFN-γ induced expression of activation markers by MDM and not AM. Human AM (**A, C, E**) isolated from bronchoalveolar lavage fluid. PBMC were isolated from buffy coats and MDM (**B, D, F**) were differentiated and adherence purified for 7 days in 10% human serum. Cells were left unprimed (black) or primed with IFN-γ (red) or IL-4 (blue) (both 10 ng/ml) for 24 hr. Cells were left untreated (solid) or treated with 2DG (5 mM; empty) 1 hr prior to stimulation with iH37Rv (MOI 1–10; square) or LPS (100 ng/ml; triangle) or left unstimulated (circle). After 24 hr, cells were detached from the plates by cooling and gentle scraping and stained for HLAR-DR (**A, B**), CD40 (**C, D**), CD86 (**E, F**) and analysed by flow cytometry. Each linked data point represents the average of technical duplicates for one individual biological donor (n=8–9). Statistically significant differences were determined using two-way ANOVA with a Tukey post-test (**A–F**); *p≤0.05, **p≤0.01, p***≤0.001, ****p≤0.001.

The online version of this article includes the following figure supplement(s) for figure 3:

**Figure supplement 1.** 2DG reduces ECAR and antigen-processing but does not reduce phagocytosis or cell viability.

with 2DG significantly reduced antigen processing in all groups, however IFN-γ primed MDM retained enhanced abilities to process antigen (*Figure 3—figure supplement 1B*). The reduced capacity of MDM to process antigen was not due to a deficiency in phagocytosis, as measured by both bead or bacterial uptake or due to increased cell death (*Figure 3—figure supplement 1C–E*).

Overall, these data suggest that IFN-γ promotes expression of activation markers via increased glycolysis in human MDM, whereas AM are not as phenotypically plastic in response to cytokine priming and subsequent stimulation. Moreover, AM upregulation of cell surface markers (with the exception of CD40) in response to priming or stimulation is not associated with glycolysis, in contrast to the MDM.

## IFN-γ enhances cytokine production in human AM more than MDM

Changes in macrophage metabolism have been previously associated with altered cytokine production (*Tannahill et al., 2013*; *Gleeson et al., 2016*; *Gleeson et al., 2018*). Having established that both IFN-γ and IL-4 can significantly alter metabolism in human macrophages we next sought to examine the ability of AM and MDM to secrete cytokines when primed with IFN-γ or IL-4. Human AM (*Figure 4A, C and E*) and MDM (*Figure 4B, D and F*) were left unprimed or primed with IFN-γ or IL-4 for 24 hr. Macrophages were then stimulated with iH37Rv or LPS. Supernatants were harvested 24 hr post stimulation and concentrations of IL-1β (*Figure 4A and B*), TNF (*Figure 4C and D*), and IL-10 (*Figure 4E and F*) were quantified by ELISA. While iH37Rv stimulation resulted in IL-1β production in unprimed AM and MDM, IFN-γ only significantly enhanced the production of IL-1β by AM (*Figure 4A and B*). IL-4 priming attenuated iH37Rv induced IL-1β in both AM and MDM (*Figure 4A and B*). As expected, IL-1β secretion was not induced in response to LPS stimulation however, in the presence of IFN-γ, IL-1β was detectable (*Figure 4A and B*). TNF was significantly induced in unprimed or IFN-γ primed, but not IL-4 primed AM in response to iH37Rv and LPS (*Figure 4C*). IFN-γ enhanced production of TNF by AM in response to both iH37Rv and LPS. In contrast, IFN-γ enhanced TNF in response to iH37Rv, but not LPS in MDM (*Figure 4C and D*). LPS significantly upregulated the production of TNF in all MDM. Notably, IFN-γ priming did not enhance TNF production and IL-4 priming significantly attenuated LPS-induced TNF (*Figure 4D*). All stimulated AM secreted IL-10 regardless of priming (*Figure 4E*). IFN-γ significantly enhanced iH37Rv induced IL-10 in AM compared to unprimed or IL-4 primed comparators (*Figure 4E*). IL-4 priming of human AM significantly reduced IL-10 production in response to iH37Rv compared with unprimed AM (*Figure 4E*). LPS strongly induced IL-10 production in unprimed MDM, which was significantly attenuated by either IFN-γ or IL-4 priming (*Figure 4F*).

These data suggest that the AM has greater functional plasticity in terms of cytokine production in response to IFN-γ than the MDM, as IFN-γ primed AM had enhanced IL-10 and TNF production, in response to Mtb or LPS, respectively. In order to directly compare the human AM and MDM responses, fold change in cytokine production was calculated compared to the average of their respective iH37Rv (*Figure 4G*) or LPS (*Figure 4H*) stimulated unprimed control. IFN-γ enhanced human AM ability to secrete IL-1β, TNF and IL-10 in response to iH37Rv compared to MDM (*Figure 4G*). The IFN-γ primed human AM also has a significantly increased ability to secrete TNF and IL-10 in response to LPS compared to MDM (*Figure 4H*); however, the difference in IL-10 secretion is more associated with MDM decreasing IL-10 when IFN-γ primed.

## IFN-γ enhanced cytokine production is markedly more reliant on glycolysis in AM compared with MDM

Since IFN-γ drove glycolysis in both AM and MDM, we next sought to examine if cytokine production was associated with enhanced glycolysis. Human AM (*Figure 5A, C and E*) and MDM (*Figure 5B, D and F*) were primed with IFN-γ or IL-4 for 24 hr or left unprimed. Macrophages were treated with 2DG (5 mM) for 1 hr prior to stimulation with iH37Rv or LPS. Supernatants were harvested 24 hr post stimulation and concentrations of IL-1β (*Figure 5A and B*), TNF (*Figure 5C and D*) and IL-10 (*Figure 5E and F*) were quantified. 2DG significantly abrogated production of IL-1β in both IFN-γ primed AM and MDM stimulated with iH37Rv (*Figure 5A and B*). Moreover, 2DG significantly reduced TNF production driven by IFN-γ in the AM, and significantly reduced TNF production in unprimed AM stimulated with iH37Rv (*Figure 5C*). Unlike IL-1β, TNF production was not affected by 2DG in unprimed or IFN-γ primed MDM. Conversely, IL-4 primed MDM exhibited increased TNF production in the presence of 2DG (*Figure 5D*). IL-10 production was significantly inhibited by 2DG in AM, irrespective of priming

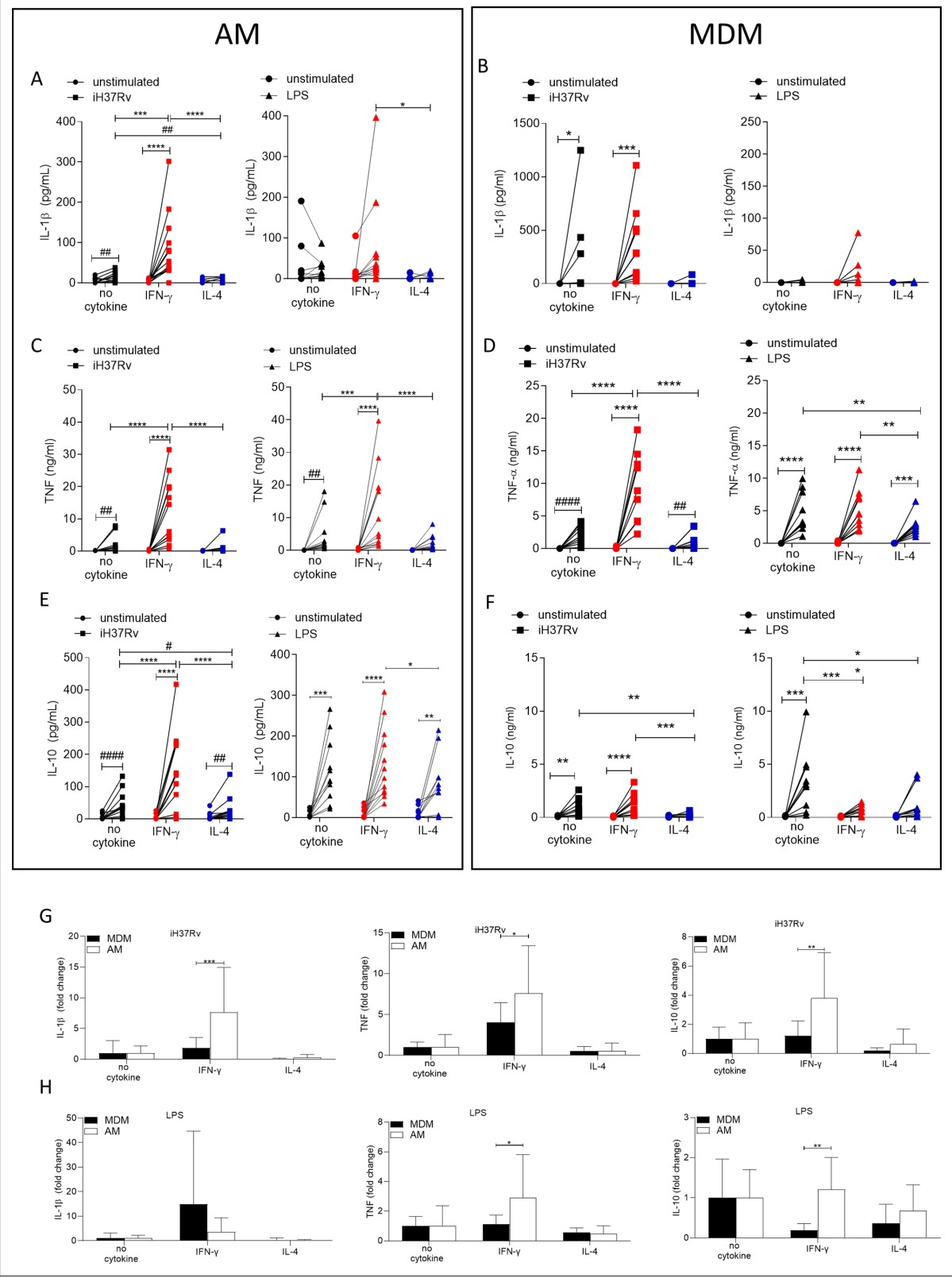

**Figure 4.** IFN-γ enhances cytokine production more in AM compared with MDM. Human AM (**A, C, E**) isolated from bronchoalveolar lavage fluid. PBMC were isolated from buffy coats and MDM (**B, D, F**) were differentiated and adherence purified for 7 days in 10% human serum. Cells were left unprimed (black) or primed with IFN-γ (red) or IL-4 (blue) (both 10 ng/ml) for 24 hr. AM or MDM were left unstimulated (circle) or stimulated iH37Rv (MOI 1–10; square) or LPS (100 ng/ml; triangle). Supernatants were harvested 24 hr after stimulation and concentrations of IL-1β (**A, B**), TNF (**C, D**), and IL-10

*Figure 4 continued on next page*

*Figure 4 continued*

(**E, F**) were quantified by ELISA. Fold change in IL-1β, TNF and IL-10 was calculated for AM and MDM based on the average of respective no cytokine controls for iH37Rv (**G**) and LPS (**H**). Each linked data point represents the average of technical duplicates for one individual biological donor (AM; n=12–13, MDM; n=8–10). Statistically significant differences were determined using two-way ANOVA with a Tukey (**A–F**) or Bonferroni post-test (**G–H**); *p≤0.05, **p≤0.01, ***p≤0.001, ****p≤0.0001 or #p≤0.05, ##p≤0.01, ####p≤0.0001 (where IFN-γ-treated data sets were excluded for post-test analysis to analyse statistical differences between no cytokine and IL-4-treated data sets).

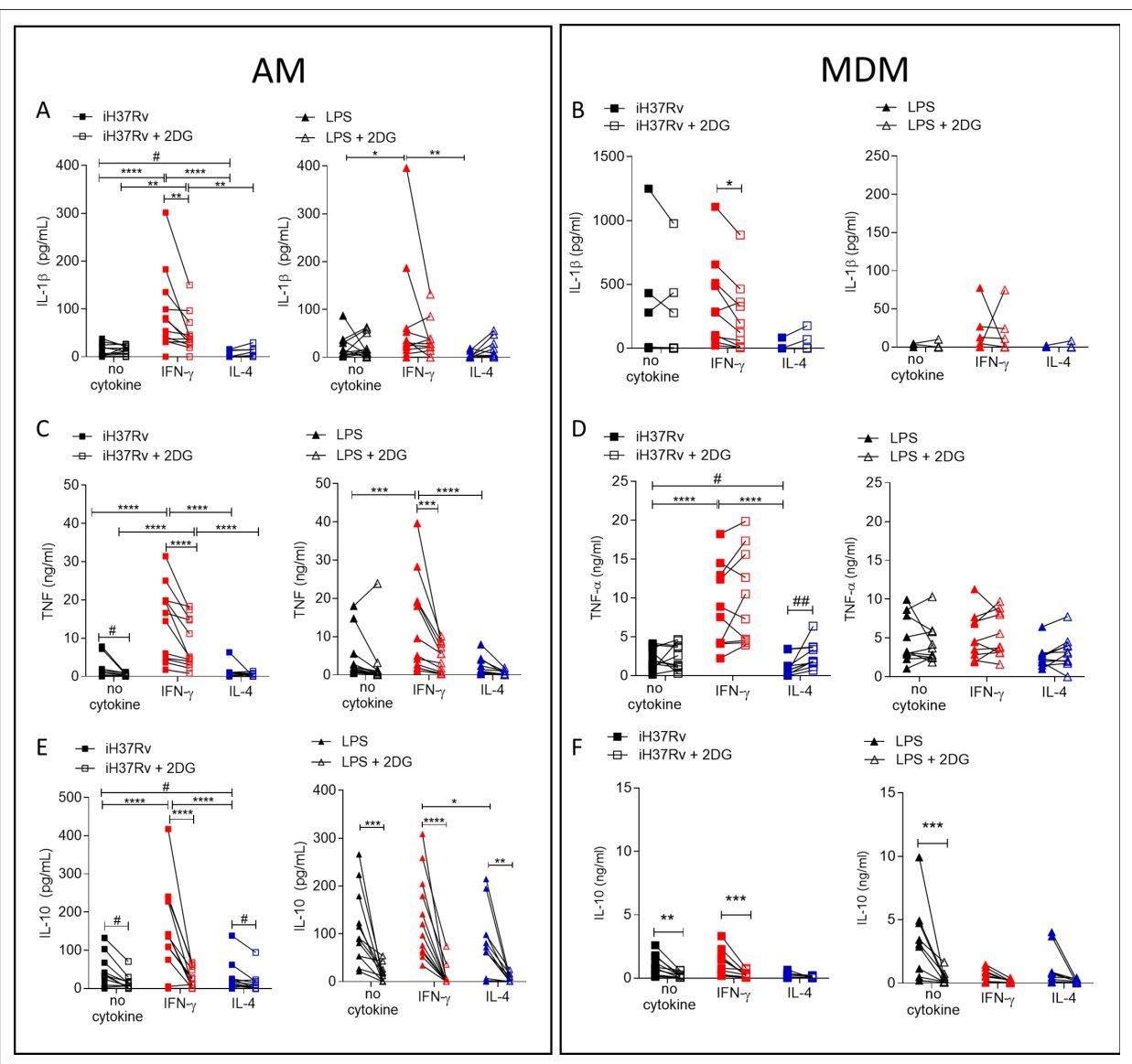

**Figure 5.** Cytokine secretion by AM is more reliant on glycolysis than MDM. Human AM (**A, C, E**) isolated from bronchoalveolar lavage fluid. PBMC were isolated from buffy coats and MDM (**B, D, F**) were differentiated and adherence purified for 7 days in 10% human serum. Cells were left unprimed (black) or primed with IFN-γ (red) or IL-4 (blue; both 10 ng/ml) for 24 hr. Cells were left untreated (solid) treated with 2DG (5 mM; empty) for 1 hr prior to stimulation with iH37Rv (MOI 1–10; square) or LPS (100 ng/ml; triangle) or left unstimulated (circle). Supernatants were harvested 24 hr after stimulation and concentrations of IL-1β (**A, B**), TNF (**C, D**) and IL-10 (**E, F**) were quantified by ELISA. Each linked data point represents the average of technical duplicates for one individual biological donor (AM; n=12–13, MDM; n=8–10). Statistically significant differences were determined using two-way ANOVA with a Tukey post-test (**A–D**); *p≤0.05, **p≤0.01, ***p≤0.001, ****p≤0.0001 or #p≤0.05, ##p≤0.01 (where IFN-γ primed data sets were excluded for post-test analysis to analyse statistical differences between no cytokine and IL-4 treated data sets).

or stimulation (*Figure 5E*). MDM production of IL-10 in response to LPS or iH37Rv was inhibited with 2DG (*Figure 5F*).

In summary, IL-1β is under the control of glycolysis in IFN-γ primed AM and MDM. TNF production is strongly under the control of glycolysis in AM but not in MDM. Cumulatively, these data indicate that IFN-γ promotes cytokine production in the AM via a process that is dependent on glycolysis. Consistent with the observation that IL-4 attenuated ECAR in LPS stimulated AM, IL-4 reduced cytokine production in the AM. Furthermore, while IL-10 was not associated with differential energetic profiles, its production is significantly attenuated by 2DG in both human macrophage populations, irrespective of priming. These data indicate that IFN-γ priming has a profound effect on AM function which is mediated, at least in part, by metabolic reprogramming.

## Discussion

AM are the first responders to infections and inflammatory insults in the lung. We and others have reported that lung resident macrophages are dependent on glycolysis to respond to LPS (*Lavrich et al., 2018*) or Mtb (*Gleeson et al., 2016*; *Gleeson et al., 2018*); however, their metabolic behaviour is distinct from murine AM (*Woods et al., 2020*) reinforcing the need to define cellular metabolism and its tractability in human macrophages in order to design effective immunometabolic therapies. We demonstrate here that human AM can be metabolically reprogrammed by IFN-γ which increased both glycolysis and oxidative phosphorylation and is further enhanced by stimulation with Mtb or LPS. This is in contrast with literature showing murine AM do not increase glycolysis in real-time following LPS stimulation (*Woods et al., 2020*), although whether IFN-γ can influence this in mice remains unknown. Our previous work shows that upon activation, the human MDM underwent 'Warburg'-like metabolism associated with an increase in glycolysis with a concomitant reduction in oxidative phosphorylation (*Ó Maoldomhnaigh et al., 2021a*). We confirmed this observation and this persists when primed with IFN-γ, unlike AM which do not undergo the shift to 'Warburg'-like metabolism. IL-4 primed macrophages are associated in the literature with an increased reliance on oxidative phosphorylation (*Van den Bossche and Saraber, 2018*). Indeed, murine IL-4 primed BMDM have increased ECAR and OCR (*Huang et al., 2016*; *Lundahl et al., 2022*). IL-4 did not promote oxidative metabolism in human AM or MDM, even when stimulated. IL-4 decreased glycolysis while preventing a decline in oxidative metabolism in MDM, inhibiting their shift to 'Warburg'-like metabolism. When directly comparing percentage change in ECAR of unprimed or macrophages primed with IFN-γ or IL-4, they all had similar rates of change despite profound differences in maximal ECAR. These data suggest that the resting state of the macrophage, and the cytokines it has recently been exposed to before activation, may be a key determining factor for the response and outcome to infections.

Directly comparing the AM with the MDM demonstrates that human AM are more reliant on oxidative metabolism upon IFN-γ priming and stimulation. Previous observations had identified a high baseline of lactate in supernatants of human AM compared to MDM (*Gleeson et al., 2016*). Interestingly, IFN-γ primed AM and MDM had significantly reduced capacity to increase ECAR (percentage change) in response to LPS stimulation, suggesting that there may be a point of maximal glycolysis. This ability to induce maximal glycolysis may be advantageous during infection as lactate, a breakdown product of glycolysis, has been shown to have antimicrobial functions against Mtb (*Ó Maoldomhnaigh et al., 2021b*; *Krishnamoorthy et al., 2020*). Moreover, pathogens such as Mtb can downregulate metabolic pathways after infection (*Cumming et al., 2018*; *Mendonca et al., 2022*) and IFN-γ is crucial for control of Mtb via glycolysis in vivo (*Braverman et al., 2016*). Based on our current data, and as recently suggested by others (*Cumming et al., 2020*), we speculate that control of Mtb in humans may be dependent on IFN-γ regulating glycolysis and not 'Warburg'-like metabolism.

To our knowledge, we are the first to demonstrate that IFN-γ alone is sufficient to cause metabolic reprogramming of both lung resident AM and peripherally derived MDM. While other studies have demonstrated a role for IFN-γ inducing metabolic alterations in macrophages these studies have focused on murine macrophages (*Wang et al., 2018*; *Braverman et al., 2016*). In contrast, evidence in murine BMDM indicates that IFN-γ alone does not increase glycolysis but that LPS was required (*Van den Bossche et al., 2015*). In addition, the use of LPS in combination with IFN-γ to polarise macrophages toward the an inflammatory phenotype is not a model easily translatable to humans, which are strikingly more sensitive to LPS than mice (*Mestas and Hughes, 2004*) and the 'M1' macrophage elicited cannot be subsequently challenged with infectious agents, as the response

is confounded by the initial LPS stimulation. Moreover, the use of LPS in addition to IFN-γ to polarise the macrophage towards the 'M1' phenotype is arguably not comparable with a macrophage that is polarised with IL-4 (or IL-10) in the absence of TLR stimuli. We wanted to assess the ability of IFN-γ alone to affect the function of human macrophages, to enable direct comparisons of macrophage subpopulations in order to fully assess the functional differences elicited in response to subsequent stimulation, in keeping with other human models (*Cumming et al., 2020*).

AM expression of activation markers was more limited compared to MDM, even when primed and stimulated. AM upregulated only HLA-DR consistently in response to IFN-γ, broadly in keeping with murine AM (*Yao et al., 2018*; *D'Agostino et al., 2020*; *Afkhami et al., 2022*). In contrast, MDM have a greater capacity to increase all activation markers in response to IFN-γ and stimulation. The response of IFN-γ primed MDM to Mtb was dependent on glycolysis for optimal activation marker expression, while LPS upregulated these markers independently of glycolysis, irrespective of cytokine priming. In contrast the AM was dependent on glycolysis for upregulation of these markers in response to LPS and not Mtb. These data once again suggest a differential role of glycolysis within human macrophages.

Glycolytic metabolism in macrophages has been intrinsically linked to cytokine production, particularly IL-1β (*Wang et al., 2018*; *Tannahill et al., 2013*; *Gleeson et al., 2016*; *Phelan et al., 2020*; *Lachmandas et al., 2016*). We have demonstrated that increased ECAR early in MDM activation is associated with increased secretion of IL-1β in both MDM and AM (*Cox et al., 2020*). Moreover, IFN-γ can promote IL-1β by inhibiting miR-21, a negative regulator of glycolysis in both human MDM and murine BMDM (*Hackett et al., 2020*). Our data builds on the observation that IFN-γ upregulates pro-IL-1β by a glycolytic-dependent mechanism in murine BMDM (*Wang et al., 2018*), by demonstrating the increased secretion of mature IL-1β by IFN-γ primed human macrophages. Moreover, we have confirmed that IL-1β secretion is dependent on glycolysis in IFN-γ primed human AM and MDM, which is line with data from murine BMDM (*Braverman et al., 2016*). In the current study, we also demonstrate that 2DG can reduce both IL-1β and IL-10 secretion by IFN-γ primed AM and MDM in response to Mtb. Moreover, 2DG inhibited LPS induced IL-10 in AM and unprimed MDM. 2DG has previously been shown to inhibit IL-10 production by LPS stimulated human MDM *Vijayan et al., 2019*; however, restricting glycolysis using glucose-free medium inhibited IL-1β but promoted IL-10 secretion (*Gleeson et al., 2016*). Furthermore, IL-10 production is inhibited by IFN-γ in human MDM, which is similar to previous data in IFN-γ primed murine BMDM where IL-10 secretion was also inhibited (*Müller et al., 2017*). However, our data also demonstrates that IFN-γ does not inhibit or promote IL-10 in human AM stimulated with LPS even though interestingly, IFN-γ primed AM had increased TNF in response to LPS. This suggests that there may be additional pathways involved in IL-10 secretion by human macrophages, which is supported by reductions in IL-10 secretion by IL-4 primed AM, which are not metabolically altered.

TNF is crucial to control infections such as Mtb (*Keane et al., 2001*; *Harris et al., 2008*; *Harris and Keane, 2010*; *Bourigault et al., 2013*). We demonstrate that IFN-γ enhances TNF production in response to Mtb stimulation in human MDM and AM; however, AM have a much greater ability to increase TNF. Moreover, IFN-γ primed AM stimulated with Mtb have significantly more production of IL-1β, TNF and IL-10 compared with unprimed controls. This is in contrast with IFN-γ primed MDM which only upregulate TNF compared to their unprimed controls. These data indicate that effective immune responses to Mtb in the lung may require AM to be primed with IFN-γ and may in part explain why patients deficient in IFN-γ or associated signalling have increased risk of TB (*Ní Cheallaigh et al., 2016*; *Remus et al., 2001*). IFN-γ-driven production of TNF is dependent on glycolysis in AM. MDM secretion of TNF is independent of glycolysis and conversely, inhibition of glycolysis in IL-4 primed Mtb stimulated MDM enhanced TNF production. Previous studies have demonstrated that glycolysis was required by murine BMDM but not AM to secrete TNF and IL-6 (*Woods et al., 2020*). Conversely, our data demonstrates that human AM need glycolysis for optimal TNF production, especially in the presence of IFN-γ, whereas MDM do not. Once again, we highlight that there is variation in the metabolic requirements within human macrophage subpopulations, and importantly, that the AM is metabolically tractable to modulate its function.

Whether the AM can respond to IL-4 has been debated (*Kulikauskaite and Wack, 2020*). Here, we demonstrate that the human AM can respond to IL-4 with evidence that IL-4 reduced glycolysis in response to LPS stimulation. In addition, AM were functionally altered by IL-4 resulting in reduced

IL-1β and IL-10 production and upregulated CD86. These data provide evidence that the human AM is capable of responding to IL-4 which may inform type 2 lung immunity, and susceptibility to infection in patients with asthma, for example.

Trained immunity improves innate responses to infection and is emerging as a key component of host directed therapies (HDT) and strategies to improve vaccine efficacy and the design of respiratory mucosal vaccines (*D'Agostino et al., 2020*; *Afkhami et al., 2022*; *Moorlag et al., 2020*; *Netea et al., 2020*; *Zhou et al., 2021*). Both IFN-γ and IL-4 can induce trained immunity in murine and human macrophages (*Yao et al., 2018*; *Lundahl et al., 2022*; *Schrijver et al., 2023*; *Li et al., 2023*). MDM trained with IFN-γ and LPS, and stimulated with Mtb had increased TNF in the acute activation phase of trained immunity (*Lundahl et al., 2022*), which we observed in IFN-γ primed MDM subsequently stimulated with Mtb. We also observed an increase in TNF, IL-1β and IL-10 in AM potentially suggesting that AM will be a target for innate training, as increased IL-1β is associated with optimal training (*Moorlag et al., 2020*; *Li et al., 2023*; *Teufel et al., 2022*). AM from mice that received an inhaled adenovirus vectored vaccine undergo trained immunity mediated by IFN-γ resulting in elevated MHC-II expression, enhanced cytokine production, and protection against specific and non-specific infection challenges (*Yao et al., 2018*; *D'Agostino et al., 2020*). We have previously demonstrated that an adenovirus vectored vaccine induces trained immunity in human monocytes and postulated that this may be IFN-γ dependent (*Murphy et al., 2023*). The current study provides evidence that IFN-γ can metabolically reprogramme the human AM, resulting in enhanced HLA-DR expression and cytokine production in response to subsequent stimulation. Cumulatively this highlights the importance of ascertaining whether IFN-γ can induce trained immunity in the human AM, which may enhance the design of respiratory mucosal vaccines.

Immune augmentation therapies delivered directly to the lung are necessary to help combat the growing threat of drug-resistant pathogens, including Mtb. We have demonstrated such approaches both in vitro and in vivo (*Cox et al., 2020*; *Phelan et al., 2020*; *Cahill et al., 2021*; *Coleman et al., 2018*; *O'Connor et al., 2019*; *Lawlor et al., 2016*). Clinical trials have indicated that nebulized IFN-γ is a viable HDT to help combat Mtb (*Dawson et al., 2009*; *Bharti et al., 2022*) and is in clinical trials for sepsis (https://clinicaltrials.gov/study/NCT04990232). Our data supports the use of inhalable IFN-γ as an immuno-supportive therapy which modulates metabolic responses. Moreover, our data indicates that IFN-γ affects metabolism and cytokine secretion in AM significantly more than MDM which lends support for the therapeutic strategy of delivering IFN-γ to the lung, by targeting the macrophage population that most need immune augmentation (*Huang et al., 2018*) and limiting potential side effects.

## Study limitations

We acknowledge that our in vitro model is simplified and may not fully reflect macrophages in vivo. Nevertheless, these data address knowledge gaps in human macrophage biology and are required to aid the translation of immunometabolism into clinical benefits in respiratory medicine. We used LPS and irradiated Mtb to model successful macrophage responses to infection. Future experiments should examine how virulent respiratory pathogens such as gram-negative *Pseudomonas aeruginosa, Klebsiella pneumoniae,* and Mtb affect human AM in Th1 or Th2 environments, to determine infection-specific effects.

The inhibition of glycolysis with 2DG cannot definitively link all observations solely to glycolysis, as limiting glycolysis will ultimately limit oxidative phosphorylation. Blocking oxidative phosphorylation with oligomycin reduced LPS induced cytokine secretion in the human AM and not MDM (*Pereverzeva et al., 2022*), both glycolysis and oxidative phosphorylation may therefore be needed for optimal AM function. The concentration of 2DG used only partially inhibited glycolysis; however, ablation of glycolysis induces significant cytotoxicity and confounds assay outcomes. Therefore, where 2DG had no effect, a role for glycolysis cannot be definitively excluded. Furthermore, only one dose of IFN-γ was utilised due to limitations in AM yield; however, recently both low and high doses of IFN-γ have been shown to have similar effects on AM in vitro (*Thiel et al., 2024*).

Establishing the immunometabolic and functional outputs of human macrophages will aid in future work examining the plasticity of the human AM. While we have established herein that the human AM is plastic in response to IFN-γ, since the AM is yolk-sac derived and long-lived, this raises the question of whether the plasticity of the AM can allow multiple sequential changes to respond and adapt to

changing microenvironments in the lung. Another question raised is whether other tissue resident macrophages behave similarly to AM or whether they have unique responses.

## Conclusion

Human AM and infiltrating MDM both increase glycolysis and oxidative phosphorylation in response to IFN-γ, and stimulation results in a further increase in glycolysis. Cumulatively, the data presented herein suggests that the MDM maybe more phenotypically plastic than the AM, while the AM have enhanced functional plasticity in their ability to produce cytokine after exposure Th1 and Th2 cytokines. Our data supports the hypothesis that there may be distinct roles for AM and infiltrating MDM during infection since IFN-γ increased metabolic responses are mechanistically associated with different cellular functions. Our data demonstrates that cytokine production in human AM can be promoted by supporting cellular metabolism, thus providing evidence that human tissue resident AM are a tractable target for host-directed immuno-supportive adjunctive therapies.

# Materials and methods

## Cell culture

Buffy coats were obtained with consent from healthy donors (aged between 18–69; ethical approval, School of Medicine Research Ethics Committee, Trinity College Dublin). Peripheral blood mononuclear cells (PBMC) were isolated by density-gradient centrifugation over Lymphoprep (StemCell Technologies). Cells were resuspended in RPMI (Gibco) supplemented with 10% AB human serum (Sigma-Aldrich) and plated onto non-treated tissue culture plates (Costar) for 7 days. Non-adherent cells were removed by washing every 2–3 days. Cultures were >90% pure based on co-expression of CD14 and CD68.

Human AM were retrieved with informed consent from patients (aged between 42–72, 50% Female) undergoing bronchoscopy, (ethical approval, St. James' Hospital (SJH) / Tallaght University Hospital (TUH) Joint Research Ethics Committee) previously as reported (*O'Leary et al., 2014*) and outlined below. Cells were plated in RPMI (Gibco) supplemented with 10% FBS (Gibco), fungizone (2.5 µg/ml; Gibco) and cefotaxime (50 µg/ml; Melford Biolaboratories). Cells were incubated for 24 hr before washing to remove non-adherent cells. Adherent cells (predominantly AM) were used for experiments.

## BAL sample acquisition

All donors were patients undergoing clinically indicated bronchoscopy and written informed consent for retrieving additional bronchial washings for research was obtained prior to the procedure. Patients were not remunerated for participation in this study. Patients were informed and consented that collective results arising from samples given would be published in academic journals. Exclusion criteria included age under 18 years, inability to provide written informed consent or a known (or ensuing) diagnosis of malignancy, sarcoidosis, HIV or Hepatitis C. Patients undergoing biopsy as part of bronchoscopy were also excluded.

Sample acquisition during bronchoscopy: Conscious sedation was achieved using intravenous midazolam and lignocaine gel was administered to the nostril. Flexible video-bronchoscope was inserted through the nostril and advanced to the level of the vocal cords by posterior approach. Further lignocaine spray was administered prior to and subsequent to traversing the vocal cords. Following routine bronchoscopy, the bronchoscope was wedged in the right middle lobe bronchus. A total of 180 ml of sterile saline was administered as 60 ml boluses via a connector inserted into the bronchoscope and aspirated within 5–10 s under low suction. The bronchoalveolar lavage fluid (BALF) was then transported directly to the laboratory for AM isolation. Pre- and post-bronchoscopy patient care was not altered by participation in the study. The procedure was prolonged by ~12 min.

## Macrophage stimulation

Macrophages were primed with IFN-γ or IL-4 (both 10 ng/ml) or left unprimed for 24 hr. Where indicated, MDM and AM were treated with 2DG (5 mM) for 1 hr prior to stimulation with irradiated Mtb strain H37Rv (iH37Rv; MOI 1–10) or LPS (100 ng/ml; Merck). For metabolic flux analysis stimulations were immediately monitored in real-time. All other stimulations were assessed after 24 hr.

## Metabolic assays

MDM were placed in ice-cold PBS and incubated at 4 °C on ice for 30 min, then gently scraped and counted using trypan blue. MDM ($1x10^5$ cells/well) were re-plated onto Seahorse plates, as previously described (*Ó Maoldomhnaigh et al., 2021a*). AM ($1x10^5$ cells/well) were directly plated onto Seahorse plates and washed after 24 hr. The ECAR and the OCR, were measured three times every 10 min to establish baselines. After 30 min, macrophages were stimulated in situ and monitored in real-time, with Seahorse medium, iH37Rv or LPS. Post stimulation the ECAR and OCR were continually sampled at 20 min intervals for times indicated. Analyses were carried out at approximately 150 min as previously described (*Ó Maoldomhnaigh et al., 2021a*). Fold change in ECAR and OCR was calculated compared with unstimulated unprimed controls at 150 min, where unstimulated unprimed macrophages were set to 1. This allows for analysis of the effects of both priming and subsequent stimulation for and accounts for the variation in the raw ECAR and OCR reading between runs thereby making each donor its own control.

Percent change in ECAR and OCR was also calculated to equalise groups to the same point prior to stimulation. Each condition was compared with its own respective primed or unprimed baseline at 30 min and this was set to 100%, prior to stimulation, this was carried to examine the capacity of cells to increase metabolic parameters in response to stimulation. Post stimulation percent change data was then extracted and analysed at 150 min. This controls for the priming effect and enables the analysis of metabolic capacity in each dataset.

## Cytokine assays

IL-1β, IL-10 (BioLegend), and TNF (Invitrogen) concentrations in supernatants were quantified by ELISA, according to manufacturer's protocol.

## Flow cytometry

Human AM and MDM were placed in ice-cold PBS and incubated at 4 °C on ice for 30 min. Cells were removed by gentle scraping, Fc blocked with Human TruStain FcX (BioLegend) and stained with zombie NIR viability dye and fluorochrome-conjugated antibodies for CD14 (Alexa488, Cat#325610), CD68 (PE, Cat#333808), CD86 (BV421, Cat#305426), CD40 (BV510, Cat#334330), and HLA-DR (APC, Cat#307610; all BioLegend). For phagocytosis and antigen processing assays, MDM were treated with fluorescent beads (Sigma-Aldrich) or DQ-Ovalbumin (Thermo Fisher) for 30 min at 37 °C, before scraping as above. DQ-Ovalbumin is fluorescent after proteasomal degradation marking antigen processing. Cells were fixed with 2% PFA and acquired on a BD FACS Canto II. Unstained and FMO controls were used to normalise for background and to set gates. Data were analysed using FlowJo.

## Statistical analysis

Statistical analyses were performed using GraphPad Prism 10. Statistically significant differences between two or more groups containing more than one variable were determined by two-way ANOVA with Tukey or Bonferroni multiple comparisons tests as stated. p-Values of ≤0.05 were considered statistically significant and denoted with an asterisk. Alternatively, p-values of ≤0.05 were denoted with a hashtag where data was analysed in the absence of IFN-γ primed data sets, to analyse statistical differences between no cytokine and IL-4-treated data sets.

## Acknowledgements

We would like to acknowledge the Irish Blood Transfusion Service for supporting our research by approving us to use anonymised un-transfused blood components for our research. We gratefully acknowledge all people undergoing bronchoscopy at St. James's Hospital Dublin who consented to take part in our research. We acknowledge the key contributions of the Clinical Research Facility at St. James's Hospital and the bronchoscopy suite, and the core facilities at the Trinity Translational Medicine Institute. The following reagent was obtained through BEI Resources, NIAID, NIH: *Mycobacterium tuberculosis*, Strain H37Rv, Gamma-Irradiated Whole Cells, NR-49098. This work was supported by The Royal City of Dublin Hospital Trust (RCDH app 185, awarded to JK), The National Children's Research Centre (D/18/1 awarded to COM) and The Health Research Board (EIA-2019–010 awarded

to SAB). Funders had no role in the study design, collection, analysis or interpretation of the data nor in the writing or submission of the article for publication.

## Additional information

### Funding

| Funder | Grant reference number | Author |
|---|---|---|
| The Royal City of Dublin Hospital Trust | RCDH app 185 | Joseph Keane |
| The National Children's Research Centre | D/18/1 | Cilian Ó Maoldomhnaigh |
| The Health Research Board | EIA-2019-010 | Sharee A Basdeo |

The funders had no role in study design, data collection and interpretation, or the decision to submit the work for publication.

### Author contributions

Donal J Cox, Conceptualization, Data curation, Formal analysis, Supervision, Validation, Investigation, Visualization, Methodology, Writing – original draft, Writing – review and editing; Sarah A Connolly, Data curation, Formal analysis, Investigation, Writing – review and editing; Cilian Ó Maoldomhnaigh, Conceptualization, Formal analysis, Funding acquisition, Investigation, Writing – review and editing; Aenea AI Brugman, Olivia Sandby Thomas, Emily Duffin, Karl M Gogan, Oisin Ó Gallchobhair, Dearbhla M Murphy, Sinead A O'Rourke, James J Phelan, Investigation, Writing – review and editing; Finbarr O'Connell, Parthiban Nadarajan, Resources, Methodology, Writing – review and editing; Laura E Gleeson, Resources, Investigation, Methodology, Writing – review and editing; Sharee A Basdeo, Conceptualization, Data curation, Formal analysis, Supervision, Funding acquisition, Investigation, Methodology, Writing – original draft, Project administration, Writing – review and editing; Joseph Keane, Conceptualization, Supervision, Funding acquisition, Methodology, Project administration, Writing – review and editing

### Author ORCIDs

Donal J Cox (ID) http://orcid.org/0000-0002-8523-3836
Sharee A Basdeo (ID) https://orcid.org/0000-0001-5616-6665

### Ethics

Human subjects: All research herein was carried out in accordance with the Declaration of Helsinki and ethically approved, as outlined in the materials and methods section by the School of Medicine Research Ethics Committee, Trinity College Dublin (REC Ref: 20200802) and the SJH/TUH Joint Research Ethics Committee (REC Ref: 2008/17/17). All individuals were provided written informed consent and were made aware that collective results arising from samples given would be published in academic journals.

Reviewer #3 (Public review): https://doi.org/10.7554/eLife.98449.4.sa1
Author response https://doi.org/10.7554/eLife.98449.4.sa2

## Additional files

### Supplementary files
• MDAR checklist

### Data availability

Datasets are available on Dryad (https://doi.org/10.5061/dryad.98sf7m0t5). No data has been omitted from this manuscript. Not all samples were used for every assay/stimulation due to limitations in cellular yield or due failure of positive or negative controls.

The following dataset was generated:

| Author(s) | Year | Dataset title | Dataset URL | Database and Identifier |
|---|---|---|---|---|
| Cox DJ, Connolly SA, Brugman AAI, Thomas OS, Duffin ED, Gogan KM, Gallchobhair OO, Murphy DM, Sinead A, O'Rourke F, O'Connell PN, Phelan JJ, Gleeson LE, Basdeo SA, Keane J | 2024 | Human airway macrophages are metabolically reprogrammed by IFN-γ resulting in glycolysis-dependent functional plasticity | https://doi.org/10.5061/dryad.98sf7m0t5 | Dryad Digital Repository, 10.5061/dryad.98sf7m0t5 |

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
