## [Editor Report · eLife Assessment]

In this **valuable** study, the authors investigate how inflammatory priming and exposure to irradiated *Mycobacterium tuberculosis* or the bacterial endotoxin LPS impact the metabolism of primary human airway macrophages and monocyte-derived macrophages. The work shows that metabolic plasticity is greater in monocyte-derived macrophages than alveolar macrophages, with **solid** experimental methods and overall evidence. The findings are relevant to the field of immunometabolism.

---

## [Referee Report · Reviewer #3 (Public review)]

Summary:

In this manuscript the authors explore the contribution of metabolism to the response of two subpopulations of macrophages to bacterial pathogens commonly encountered in the human lung, as well as the influence of priming signals typically produced at a site of inflammation. The two subpopulations are resident airway macrophages (AM) isolated via bronchoalveolar lavage and monocyte-derived macrophages (MDM) isolated from human blood and differentiated using human serum. The two cell types were primed using IFNγ and Il-4, which are produced at sites of inflammation as part of initiation and resolution of inflammation respectively, followed by stimulation with either heat-killed tuberculosis (Mtb) or LPS to simulate interaction with a bacterial pathogen that is either gram-negative in the case of Mtb or gram-positive in the case of LPS. The authors use human cells for this work, which makes use of widely reported and thoroughly described priming signals, as well as model antigens. This makes the observations on the functional response of these two subpopulations relevant to human health and disease to a greater extent that the mouse models typically used to interrogate these interactions. To examine the relationship between metabolism and functional response, the authors measure rates of oxidative phosphorylation and glycolysis under baseline conditions, primed using IFNγ or IL-4, and primed and stimulated with Mtb or LPS.

Overall, this study reveals how inflammatory and anti-inflammatory cytokine priming contributes to the metabolic reprogramming of AM and MDM populations. Their conclusions regarding the relationship between cytokine secretion and inflammatory molecule expression in response to bacterial stimuli are supported by the data. The involvement of metabolism in innate immune cell function is relevant when devising treatment strategies that target the innate immune response during infection. The data presented in this paper further our understanding of that relationship and advance the field of innate immune cell biology.

---

## [Author Response]

The following is the authors’ response to the previous reviews.

**Reviewer #2 (Recommendations for the authors):**
Comments to the authors:R1. The authors show a similar reduction in ECAR as a measure of glycolytic inhibition upon treating H37Rv-infected unprimed MDMs with 5 mM 2-DG at 1 h and 24 h. However, the data pertaining to the extent of glycolytic inhibition upon 2-DG treatment in IFN-γ or IL-4 primed AMs or MDMs is not included.

We acknowledge that we have not checked the ECAR of every dataset herein treated with 2DG. However, we have provided evidence that 2DG reduced ECAR in the control datasets, and moreover, 2DG is functionally affecting the cells (e.g. the presence of 2DG altered cytokine production in both AM and MDM, even in the presence of IFN-γ or IL-4).

R2. The authors have replotted the same data as percent change and fold difference with different normalizing samples. While they have corrected one of the highlighted discrepancies in the data plotting of Fig. 1A and 1C, similar discrepancies are still evident in many other instances. Based on my understanding of the data and normalization methodology stated by the authors in response to comment (#5) by reviewer 1, the authors are plotting fold changes across all samples with respect to unstimulated and unprimed macrophages, whereas percent changes are plotted for stimulated (LPS or dead H37Rv) samples with respect to baseline measurements for each unstimulated sample under differently primed macrophages. I believe the slope of lines connecting unstimulated and LPS stimulates/H37Rv infected upon percent increase or decrease (from the baseline of unstimulated samples) will still maintain their trend in fold changes (relative to unstimulated and unprimed macrophages) irrespective of changes in absolute values. For instance, in Fig. 1F, there are at least 3 samples that show an increase in fold change in OCR upon H37Rv infection in IFN-γ primed MDMs. However, Fig. 1H, plotted from the same data, shows a decrease in OCR across all IFN-γ primed MDMs upon H37Rv infection. The authors have also highlighted that this decrease in OCR upon H37Rv infection in IFN- γ primed MDMs is highly significant (P < 0.01). The same data is again plotted as a bar plot in Fig. 1J as fold change relative to unstimulated and unprimed macrophages (mislabeled as percent change to unstimulated), showing no difference upon H37Rv infection of IFN-γ primed MDMs.

We have amended the axis in Figure 1 and Supplemental Figure 1 to more accurately describe the two different forms of analysis. We have fixed the errors outlined. We have also amended the methods in the text to clarify the two analyses carried out on the metabolic data. Lines 113-122 as follows:

“Fold change in ECAR and OCR was calculated compared to unstimulated unprimed controls at 150 minutes, where unstimulated unprimed macrophages were set to 1. This allows for analysis of the effects of both priming and subsequent stimulation for and accounts for the variation in the raw ECAR and OCR reading between runs thereby making each donor its own control.

Percent change in ECAR and OCR was also calculated to equalise groups to the same point prior to stimulation. Each condition was compared to its own respective primed or unprimed baseline at 30 minutes and this was set to 100%, prior to stimulation, this was carried to examine the capacity of cells to increase metabolic parameters in response to stimulation. Post stimulation percent change data was then extracted and analysed at 150 minutes. This controls for the priming effect and enables the analysis of metabolic capacity in each dataset.”

For figure 1J, the data is replotted from fold change datasets (not percentage change where the decrease in OCR is significant). The axis label has been revised for accuracy.